# Adherence to Medication in Neurogeriatric Patients: Insights from the NeuroGerAd Study

**DOI:** 10.3390/jcm11185353

**Published:** 2022-09-13

**Authors:** Aline Schönenberg, Hannah M. Mühlhammer, Thomas Lehmann, Tino Prell

**Affiliations:** 1Department of Geriatrics, Halle University Hospital, 06120 Halle, Germany; 2Department of Neurology, Jena University Hospital, 07747 Jena, Germany; 3Institute for Medical Statistics, Computer and Data Sciences, Jena University Hospital, 07747 Jena, Germany

**Keywords:** depression, older adults, medication adherence, quality of life, multimorbidity

## Abstract

Nonadherence to medication is associated with increased morbidity, mortality, and healthcare costs, especially in older adults with higher chances of multimorbidity. However, comprehensive data on factors influencing adherence in this patient group are rare. Thus, data for 910 patients were acquired, including demographic data, nonadherence (Stendal Adherence to Medication), depression (Beck Depression Inventory), cognition (Montreal Cognitive Assessment), personality (Big Five Inventory), satisfaction with healthcare (Health Care Climate Questionnaire), quality of life (36-item Short Form Survey), mobility, diagnoses, and medication. Elastic net regularization was used to analyze the predictors of adherence. Principal component and general estimation equations were calculated to analyze the underlying patterns of adherence. Only 21.1% of patients were fully adherent. Nonadherence was associated with male gender, higher number of medications, diagnosis, depression, poor patient–physician relationship, personality, impaired cognition, and impaired mobility. Nonadherence was classified into three sub-factors: forgetting (46.2%), missing knowledge about medication (29%), and intentional modification of medication (24.8%). While depression exerted the strongest influence on modification, a high number of medications was associated with missing knowledge. The different patterns of nonadherence (i.e., modification, missing knowledge, and forgetting) are influenced differently by clinical factors, indicating that specific approaches are needed for interventions targeting adherence.

## 1. Introduction

The treatment of chronic disorders commonly includes the long-term use of pharmacotherapy, and older adults especially are often expected to adhere to complex drug regimes [1]. Adherence is described as the extent to which a person’s behavior corresponds to the recommendations from their healthcare providers [2]. However, many older adults either cannot, or do not want to, take medications as prescribed [3]. This nonadherence to medication contributes to adverse drug events, increased length of stay and readmissions to hospitals, higher healthcare costs, lower quality of life (QoL), and poorer health outcomes [2,4,5,6]. In general, nonadherence may be intentional, i.e., when a patient purposefully decides not to follow the recommended treatment, or unintentional, meaning that a patient cannot follow the recommendations, for example due to cognitive or physical impairments [7]. Several factors are known to contribute to nonadherence, such as depression and cognition [8]. While multiple studies have been conducted on the predictors of nonadherence in specific illnesses (e.g., hypertension, COPD, asthma, HIV, etc.), little is known about the mechanisms of nonadherence in elderly patients with neurological disorders [9], despite the fact that over 20% of adults aged 60 and older have a mental or neurological disorder [10]. As nonadherence poses problems for both patients and healthcare systems, it is essential to investigate further the occurrence of medication nonadherence and associated factors in this growing cohort.

We collected comprehensive data on adherence and its modifying factors from geriatric patients with neurological disorders. Additionally, we sought to understand whether adherence is influenced not only by known predictors (e.g., depression), but also by the underlying neurological disease itself [11]. Furthermore, we aimed to determine whether different patterns of nonadherence (i.e., intentional and unintentional) are influenced differently by clinical parameters.

## 2. Materials and Methods

### 2.1. Settings and Participants 

This study was registered in the German Clinical Trials Register (registration number: DRKS00016774; registered on 2 February 2019), and the study protocol was published in advance [11]. The study was approved by the local ethics committee (approval number: 5290-10/17) of Jena University Hospital. All patients provided written informed consent. From February 2019 to March 2020, elderly patients with neurological disorders received a comprehensive geriatric assessment during their stay in the Department of Neurology. This study reports the results of the cross-sectional assessments. 

We included patients (age > 60 with multimorbidity or age > 70) with a common neurological disorder (e.g., cerebrovascular disorders, movement disorders, epilepsy, and neuromuscular or peripheral neurological disorders). Patients with dementia, acute psychotic symptoms, or delirium were excluded. We screened all patients in the Department of Neurology for eligibility. Among the 2021 patients aged 60 years or older admitted during data collection, 113 were missed for timing reasons. Of the remaining 1908 patients, 997 were excluded because they did not meet the inclusion criteria, declined to participate, or were prevented from participating due to other medical reasons (e.g., unconsciousness or inability to speak). In total, 995 patients were eligible, of which 910 patients participated in the study. Thus, data for 910 patients were analyzed. A description of the screening procedure is provided in Appendix A Appendix A.

Individual deviations from the study protocol needed to be made due to the onset of COVID-19 in the last three months of data collection. This resulted in a drastic reduction in the ward occupancy rate and the desired sample size of 250 subjects per neurological disorder was not reached. This limits the significance of the findings, especially for patients with epilepsy; thus, no conclusive statements can be made here. Additionally, we included patients younger than 60 years with multimorbidity (*n* = 139, aged between 55 and 59 years).

### 2.2. Assessments 

The paper reports cross-sectional results on overarching factors influencing nonadherence in our population of older patients. Therefore, the primary outcome variable was nonadherence according to the Stendal Adherence to Medication Score (SAMS) [12]. Briefly, the SAMS is a self-report questionnaire consisting of 18 items, scores for which are totaled to produce a cumulative adherence score, with 0 indicating complete adherence and 72 complete nonadherence. The items are rated on a 5-point Likert scale. Scores for different sub-factors can be calculated, namely, for *forgetting* to take medication, intentional *modification* of medication, and *missing knowledge* about medication. *Modification* refers to the adjustment of medication (dosage, time points) without consulting a doctor, while *missing knowledge* represents patients who were unaware of the purpose of their medication and/or dosages. The factor *forgetfulness* includes patients who unintentionally forget to take their medication [12,13,14]. 

Patients’ cognitive ability was tested using the Montreal Cognitive Assessment (MoCA) [15]. The MoCA result, along with the clinical impression during the face-to-face screening procedure, allowed us to decide whether the patient was able to provide valid self-assessments and could be included or not.

The following variables were recorded from the patients’ medical records: age, gender, main neurological diagnosis, and medication regimen at admission and discharge.

The following variables were recorded via self-report: marital status (single, divorced, widowed, or married); living condition (alone or not alone); level of education (high: German abitur or university; medium: German Realschule or general certificate of secondary education; or low: German Hauptschule or no school); employment status; number of medications per day; medical diagnoses; Beck Depression Inventory (BDI) score [16]; Big Five Inventory (BFI) scores [17]; Health Care Climate Questionnaire (HCCQ) scores [18]; Stendal Adherence to Medication Score (SAMS) [12] and results of the 36-item Short Form Survey (SF-36) to measure QoL [19].

The following variables were recorded in face-to-face assessments by trained study staff: changes in medications in the last six months (yes/no); Timed Up and Go test (TuG) [20], if medically possible; MoCA; use of walking aids; use of visual aids; use of other aids; regular physiotherapy (yes/no); occupational therapy (yes/no); speech therapy (yes/no); and frequency of neurologist/GP consultations. See Appendix A Appendix A for details of the questionnaires. 

### 2.3. Statistical Analysis

Descriptive statistics were used to describe the study population. Mean and standard deviation (SD) are reported for continuous variables, and categorical variables are presented as absolute and relative frequencies. Missing data were treated according to the pairwise deletion process [21].

As a first step, linear regression with elastic net regularization was performed to determine the predictors of the total SAMS score [22]. Elastic net regularization performs variable selection by shrinking the parameters toward zero and attenuating overfitting, a well-known problem when applying regression models [23], and leads to interpretable, parsimonious models. Tenfold cross validation was performed to choose the model with the lowest mean cross-validated error. Within the elastic net algorithm, variables remain in the model if the prediction error averaged over the cross-validation samples is reduced. In contrast to ordinary least squares regression, or least absolute shrinkage and selection operator regularization, the elastic net algorithm performs well with highly correlated variables, either including all variables with similar regression coefficients or excluding all variables from the best model. Regression coefficients of the model with 95% confidence intervals (CIs) were reported. Elastic net regularization was performed using the package glmnet [24] in R version 3.6.2 (R Foundation for Statistical Computing, Vienna, Austria).

A principal component analysis (PCA) with varimax rotation was performed to assess the underlying structure of the adherence (SAMS) data and to confirm the three factors found in previous literature [12,13,25].

Subsequently, to understand the predictors of adherence in more detail, generalized estimating equations (GEEs) [26] were developed to assess the influence of the factors (gender, education, living situation, diagnosis group, and BFI) and covariates (age, medication intensity, TUG, and MoCA) on the different SAMS sub-factors *modification*, *missing knowledge*, and *forgetting*. Since three-factor scores for each patient were evaluated in one model, the correlation of these measurements required to be considered; therefore, GEE models for correlated data were fitted following the steps described below [27]:(i)Fit a standard regression model assuming that observations are independent(ii)Take the residuals from the regression and use them to estimate the parameters that quantify the correlation between observations in the same individual.(iii)Refit the regression model using a modified algorithm incorporating a matrix that reflects the magnitude of the correlation estimated in step ii.(iv)Keep alternating between steps ii and iii until the estimates stabilize.

An exchangeable covariance structure was used assuming that every observation (i.e., factor score) of a patient was equally correlated with the other factor scores of that patient. Robust standard errors were calculated to ensure consistent inferences from a GEE model even if the prespecified covariance structure was inappropriate.

All statistical tests were applied two-sided at a significance level of 0.05.

## 3. Results

Nine hundred and ten adults participated in the study, consisting of 389 female and 521 male patients aged 70 ± 8.6 years. The main neurological diagnoses derived from the patients’ medical records were movement disorders (*n* = 303; 33.3%), cerebrovascular disorders (*n* = 233; 25.6%), neuromuscular and peripheral neurological disorders (*n* = 168; 18.5%), epilepsy (*n* = 48; 5.3%) and miscellaneous diagnoses (*n* = 158; 17.4%). The characteristics of the cohort are summarized in Table 1 and the mean levels for the eight SF-36 subscales are presented in Figure 1, showing that QoL was substantially impaired in our patients compared to the general German population as assessed by the German Health Interview and Examination Survey for Adults [28].

The distribution of the SAMS results is given in Figure 2.

Initially, PCA was used to reduce the 18 SAMS items into three factors representing different reasons for nonadherence (see Appendix A for the item classification). According to our previous research [13], we attributed these three factors to *modifications*, *missing knowledge*, and *forgetfulness*. For every patient with a SAMS > 1 point (*n* = 608), the regression coefficients for each PCA factor were calculated; the highest value indicated into which group the patient was categorized: 281 (46.2%) belonged to the *forgetting* group, 176 (29.0%) to the *missing knowledge* group, and 151 (24.8%) to the *modification* group.

As an initial step to understand overall adherence, elastic net regularization was applied to determine the predictors for the total SAMS score. Increased adherence was associated with female gender (*p* < 0.001), whereas nonadherence was associated with higher levels of depression (*p* < 0.001), lower HCCQ scores (*p* = 0.03), and impaired mobility (*p* = 0.01) (Table 2).

To understand the predictors of adherence in more detail, additional models were calculated to determine the predictors of the SAMS sub-factors (Table 2). Our analyses revealed that *modification* of medication was significantly increased by depression (*p* < 0.001), but reduced by a higher number of daily medications (*p* < 0.001) and neurotic personality traits (*p* = 0.02). *Forgetting* to take medication was enhanced by living with a partner (*p* = 0.02), depressive symptoms (*p* = 0.03) and additional use of non-medical treatment (*p* = 0.04). In contrast, female gender (*p* = 0.04), low education (*p* = 0.03) and a neuromuscular disorder as main diagnosis (*p* = 0.01) decreased the probability of *forgetting* to take medication. Finally, *missing knowledge* was associated with higher age (*p* = 0.01), male gender (*p* = 0.03), worse cognitive performance (*p* < 0.001), higher levels of depressive symptoms (*p* = 0.01) and an increasing number of daily medications (*p* < 0.001).

Lastly, as depression is a known predictor of nonadherence and was related to all SAMS factors in our analysis, we aimed to answer exactly how the different SAMS factors are influenced by depression using a GEE model (Table 3). We found significant main effects for gender (*p* = 0.001) and depression (*p* = 0.039) and additionally observed significant interactions for modification with the number of medications (*p* = 0.001) and depression (*p* = 0.017), as well as for missing knowledge with number of medications (*p* = 0.013) and MoCA (*p* < 0.001). In the univariate regression models for each SAMS factor, we again found that the number of medications per day (*p* < 0.001) and depression (*p* < 0.001) exerted the strongest influence on *modification*, whereas the number of medications (*p* < 0.001), MoCA (*p* < 0.001) and depression (*p* = 0.042) had the strongest impact on *missing knowledge*. *Forgetting* was enhanced by depression (*p* = 0.057) and decreased by living alone (*p* = 0.03) (Appendix A).

## 4. Discussion

This cross-sectional study examined the predictors of self-reported nonadherence in hospitalized older patients with neurological diseases. Sociodemographic variables, personality, depression, cognition, mobility, and satisfaction with healthcare providers were related to adherence, which conforms to the findings of other studies [6,8,9,31,32]. Furthermore, although depression and number of medications remained influential in all analyses, the different subfactors of nonadherence were influenced differently by the parameters considered. This is of enormous importance for developing interventions to improve adherence. The results and methodological features of the study are discussed below.

According to the results obtained for the SF-36, the cohort studied showed poorer QoL in all domains compared with a German reference cohort, the German Health Interview and Examination Survey for Adults (DEGS1), confirming that having one or more chronic diseases was associated with lower values in all QoL domains [28]. The largest difference between our cohort and the reference cohort was observed for physical function and role limitations due to physical problems. This finding is mirrored in other studies linking multimorbidity or chronic illness to worse functional status, disability, and reduced QoL [33,34].

This study revealed several predictors of global nonadherence and different types of nonadherence, which can broadly be divided into patient factors, interpersonal factors, and medication factors [6]. As in our previous work, we used the SAMS to detect *modification*, *missing knowledge*, and *forgetting* to take medication [13,14,25]. These factors were influenced differently by clinical and demographic variables.

The main patient factors associated with changes in adherence were depression, gender and cognitive function. This conforms to many other adherence studies in older adults [6]. Mirroring the literature, depression was identified as one of the main factors influencing adherence for all domains. Interestingly, in this study, depression was most closely linked to *modification*. One possible explanation for the effect on modification in depressed patients may be the reduced belief in the efficacy of medication, as depression is associated with reduced self-efficacy and patients may no longer believe in their ability to influence their illness [35,36,37]. A failure to perceive the benefit of medication, a general perception of illness, and illness burden reduce adherence, all of which depressed patients may be more sensitive to [6,36,38].

Furthermore, higher cognitive ability was associated with higher adherence in the missing knowledge category, as it is easier for cognitively unimpaired patients to understand and remember information about medication. Similarly, increased age was associated with more missing knowledge. These results conform to those found in the existing literature, highlighting the effect of cognitive impairments on reducing adherence [39,40].

Regarding the influence of age, previous studies have reported differing results, but, often, increased age is found to be detrimental to adherence due to its relation with cognitive decline [6]. This interpretation is supported by our results, which showed that increase in age was associated with reduced adherence, especially for the missing knowledge subfactor, which was also influenced by cognition. Of note, the influence of age on nonadherence has been found to be most pronounced when studies include participants that span a wide age range, as advanced age is associated with declines in cognition and health, with older patients differing strongly from their younger counterparts [41]. In our analysis, the selective inclusion of only patients of advanced age potentially resulted in reduced influence of age as a predictor.

In contrast to studies showing that neuroticism is associated with reduced adherence [6,39], in our study, neuroticism was associated with increased adherence for the *modification* group. A possible explanation is that other studies did not differentiate between different types of nonadherence, and neurotic patients may be too afraid to willfully change their medications without consulting their doctor.

Interestingly, we found gender differences, with women reporting better adherence than men, especially in the *missing knowledge* group. There are mixed results in the literature regarding sex differences in adherence [40], although most studies have not reported differences [37]. Further studies are needed to understand where these differences stem from and how they can be overcome.

Education is often cited as an influential factor for nonadherence [6,14] and our data confirmed this. Lower education was associated with nonadherence in the *missing knowledge* group, and, interestingly, it decreased the chances of *forgetting* medication. Patients with lower education may be more careful with their medication if they do not feel equipped to deal with possible complications or worsening of symptoms, for various reasons, spanning both cognition and socioeconomic status. Although education is often discussed as an intervention method for increasing adherence [2,9], it is important to keep in mind that the education level measured in this study was not medication-specific.

Regarding interpersonal factors, we found that trust in health care providers was a predictor of increased adherence [6,42]. Similarly, living alone was associated with better adherence. This was also observed in an early study of hypertensive patients [43]. However, according to another study on older adults, living alone was associated with lower adherence, although this study focused on cognitively impaired patients [44]. Since the majority of our patients evidenced normal cognition, it is possible that, for them, living alone and being solely responsible for their health led to more accountability and thus higher adherence.

In terms of medication factors, reports in the literature suggest increased nonadherence when patients take more medications or report frequent changes [6,45]. Furthermore, it is important to keep in mind that the number of medications per day is also an indicator of multimorbidity, and therefore of worse health in general. Several studies have highlighted the connection between nonadherence and the number of medications or the complexity of the medication regime [46,47]. Our analyses showed that the number of medications was primarily related to *modification* and *missing knowledge*; thus, we argue that such complex medication plans are either too complicated for patients to understand or are accompanied by adverse side-effects leading to nonadherence [38,46]. This idea is supported by studies showing that education on medication can improve knowledge and adherence [48], and that simpler dosing regimens lead to increased adherence [47]. Interestingly, our analyses revealed reverse effects for patients in the *modification* group, where an increased number of medications reduced nonadherence. One possible explanation for this seemingly contradictory finding is that patients no longer dare to modify their medication regime when it becomes too complex, for fear of interfering with the intricate interplay of different agents.

Another interesting medication-related factor is the use of non-medical treatments, such as physiotherapy, which was revealed as a relevant factor increasing nonadherence in the *forgetting* group but not in the other groups. This seemingly contradictory finding may be explained by a busier schedule which may lead to forgetting medication before or after therapy sessions. Alternatively, patients may place less value on pharmacological treatment when also using nonpharmacological approaches, thus forgetting to take their medication often enough. However, our data does not allow for any explanation of this finding and further studies are needed to analyze the relationship between pharmacological and nonpharmacological treatments.

Our original hypothesis was that underlying neurological disorder impacts adherence [11]. Our data partially support this hypothesis, as diagnosis was a relevant factor in the elastic net model, especially for neuromuscular disorders. Of note, the listed diagnoses were not mutually exclusive, as many older patients suffer from multiple illnesses and may therefore share underlying diagnoses [34], which may effectively eliminate differences caused by individual diagnoses. For example, a patient diagnosed with Parkinson’s disease may also previously have suffered a stroke, thus sharing characteristics with patients classified as ‘cerebrovascular’ in our dataset. The diagnoses listed in our data represent the most recent main diagnoses that patients were treated for at the time of recruitment; however, due to the presence of secondary diagnoses, this classification is not conclusive, which may explain the lack of support for our hypothesis. Due to the high occurrence of multimorbidity in the older population [34], it is rarely possible to find patients suffering exclusively from one health issue; this complexity should be taken into account when undertaking research on this patient population.

To summarize these complex results, our findings mirror the previous literature in highlighting the detrimental influence of depressive symptoms on adherence across all subfactors [6]. We were also able to confirm the number of medications as an influential factor [47], although our data suggest a differential influence on certain sub-factors of nonadherence, with a higher number of medications potentially protecting against intentional *modification* of medication. Other influential parameters, such as cognition, education and gender, mainly influenced *missing knowledge* and *forgetting*, with female gender increasing adherence for both subfactors. Cognitive deficits were most closely linked to *missing knowledge* but not *forgetting* of medication [6].

### 4.1. Limitations

This study has several limitations. Although the observed predictors and prevalence of nonadherence are comparable to other studies, the results are restricted to hospitalized neurogeriatric patients. As we were interested in personal factors, we used self-reports to assess nonadherence. Although this is a common and legitimate approach [49], it does not allow for statements to be made about the actual medication adherence ratio or the correctness of drug intake. Furthermore, we also collected other information through self-reports, which are prone to biases [50]. However, all the questionnaires used are widely reported in the clinical literature and have been validated. Although we have collected a large amount of clinical data, capturing all relevant factors is inevitably not possible. In our opinion, significantly increasing the number of assessments made of this patient group risks creating datasets that are incomplete or invalid, as older adults grow tired or lose focus. As mentioned above, there were some necessary adaptations made to the study protocol because of the COVID-19 pandemic. Furthermore, the prevalence of nonadherence mainly depends on the threshold used to determine nonadherence [51]. In many studies using electronic pill monitoring or the medication possession ratio, a value of 20–25% is commonly regarded as the threshold for clinically relevant nonadherence. In addition, for self-reported adherence, several cutoff values have been used in previous studies [14,52]. According to the SAMS, using one point as an indicator of nonadherence, 78.9% of the screened patients reported some degree of nonadherence. However, not every degree of nonadherence is clinically relevant, and the threshold value at which nonadherence becomes clinically relevant has not yet been sufficiently investigated [51]. For the cohort studied in this study, there is no clear external criterion against which the effect of nonadherence can be measured, such as blood pressure during antihypertensive therapy. Therefore, we did not use a cutoff value for the SAMS but instead used it as a continuous variable.

### 4.2. Conclusions

Overall, the aim of our analysis was to detect factors pertaining to nonadherence to medication in geriatric patients with neurological disorders, with a special focus on different subfactors of nonadherence. Our data suggest a complex interplay of various factors relating to nonadherence, with depression and the number of medications being the most influential parameters. Highly complex medication regimes may lead to nonadherence, especially due to missing knowledge, but, at the same time, a higher number of medications reduces the chance of patients intentionally modifying their medication. Depression increases the chances of nonadherence across all subfactors. Therefore, both depressive symptoms and the complexity of medication should be targeted in interventions to assist patients with their medication. In addition, our results highlight the need to differentiate between different types of nonadherence, as other influential parameters, such as cognition or gender, influence different adherence subfactors to varying degrees. These results once more highlight the complexity of adherence and underline the necessity of assessing individual reasons for nonadherence to provide patients with the most effective support.

## Figures and Tables

**Figure 1 jcm-11-05353-f001:**
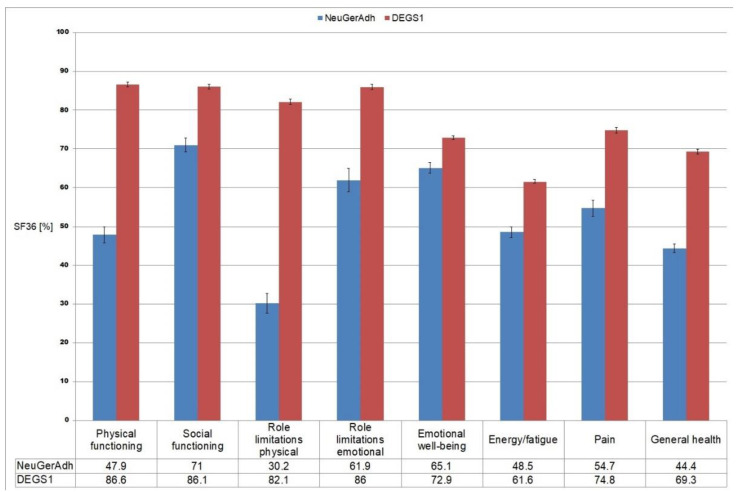
Mean and 95% Confidence intervals from SF-36 domains in comparison with the general German population. Note: NeuGerAdh = Data from the current study, DEGS1 = German Health Interview and Examination Survey for Adults.

**Figure 2 jcm-11-05353-f002:**
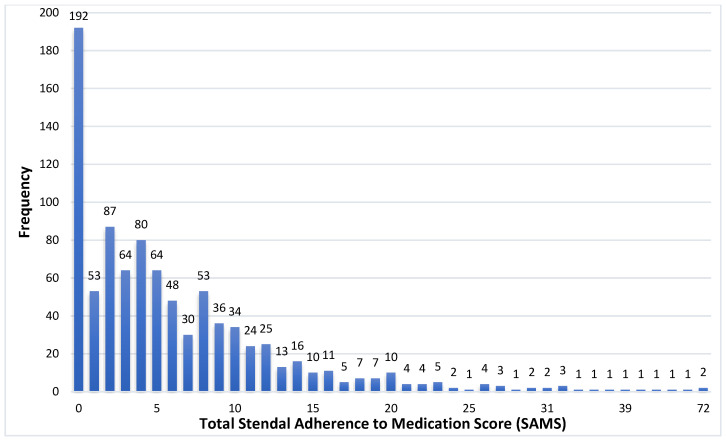
Histogram of the Stendal Adherence to Medication Score (SAMS).

**Table 1 jcm-11-05353-t001:** Clinical and demographical characteristics.

Variable	Value	*n*	%
Sex	Female	389	42.7
	Male	521	57.3
Marital status	Single/widowed/divorced	277	30.8
	Married	621	69.2
Living situation	Alone	204	24.1
	Not alone	641	75.9
Education	High	325	36.3
	Middle	306	34.2
	Low	265	29.6
Occupation status	No work	756	84.0
	Working	144	16.0
Diagnosis group	Movement disorder	303	33.3
	Cerebrovascular disorder	233	25.6
	Epilepsy	48	5.3
	Neuromuscular	168	18.5
	Others	158	17.4
Depression according to BDI [16]	No depression	468	51.4
	Minimal depression	187	20.5
	Mild depression	139	15.3
	Moderate depression	61	6.7
	Severe depression	27	3.0
Cognition [29]	Normal (MoCA ≥ 23)	536	61.1
	deficits (MoCA < 23)	300	35.9
Mobility (TuG) [20]	1–20 s	558	61.3
	20–30 s	22	2.4
	>30 s	5	0.5
Use of walking aids	Yes	297	32.6
	No	547	60.1
Use of visual aids	Yes	596	65.5
	No	247	27.1
Use of other aids physiotherapy	Yes	221	24.3
	Yes	356	39.1
	No	488	53.6
Occupational therapy	Yes	125	13.7
	No	719	79.0
Speech therapy	Yes	57	6.3
	No	787	92.7
Medication change in the last 6 months [30]	Yes	387	45.9
	No	457	54.1
Medication preparation	Independent	706	77.6
	Needing help	141	16.6
Adherence	Total Adherence (SAMS = 0)	192	21.1
		**M**	**SD**
Age		70.1	8.6
BDI sum score		9.8	7.6
HCCQ		5.6	1.1
MoCA		22.5	4.4
SAMS		6.3	7.6
TuG duration in seconds		10.5	4.3
Quarterly frequency of consultation with neurologist (or GP if neurologist is not available)		2.1	2.7
Number of medications per day (Range: 20–0)		5.6	3.6

Note: BDI = Beck Depression Inventory, HCCQ = Healthcare Climate Questionnaire, MoCA = Montreal Cognitive Assessment, GP = General Practitioner, SAMS = Stendal Adherence to Medication Score, TuG = Timed Up and Go Test.

**Table 2 jcm-11-05353-t002:** Summary of elastic net regularization for SAMS and SAMS factors.

Input Variables	SAMS Total		Modification		Missing Knowledge		Forgetting	
	Coeffic.	*p*	Coeffic.	*p*	Coeffic.	*p*	Coeffic.	*p*
Age			−0.01	0.24	0.01	0.02		
Gender: female	−1.85	<0.001			−0.16	0.03	−0.18	0.04
Education: Middle Low	−0.35	0.56			−0.05 0.17	0.58 0.07	−0.21	0.03
Living: not alone	0.56	0.40					0.24	0.02
Number of medications/day			−0.04	<0.001	0.05	<0.001	0.01	0.21
Diagnosis group: Cerebrovascular * Epilepsy * Neuromuscular * Other *	−1.09 −1.23 −1.28	0.42 0.09 0.09	−0.27 0.12	0.19 0.26	0.19 −0.10	0.28 0.29	−0.25	0.01
BDI	0.31	<0.001	0.04	<0.001	0.01	0.01	0.01	0.03
HCCQ	−0.57	0.03	−0.04	0.30			−0.07	0.09
BFI Conscientiousness + Neuroticism + Openness + Agreeableness +	−1.27 0.83	0.17 0.44	−0.34	0.02				
MoCA	−0.07	0.50	0.02	0.33	−0.07	<0.001		
TuG	0.13	0.05	0.01	0.19	0.01	0.14		
Use of non-medical treatment	0.18	0.76					0.18	0.04
Change of medication in last 6 months					0.07	0.38		

* in reference to Parkinson’s disease, + in reference to extraversion. Note: cells are left blank if the respective variable was no longer included in the final model after variable selection via elastic net regularization. SAMS: Stendal Adherence to Medication Score, BDI II: Beck Depression Inventory II, HCCQ: Health Care Climate Questionnaire, BFI: Big Five Inventory, MoCA: Montreal Cognitive Assessment, TuG: Timed Up and Go test.

**Table 3 jcm-11-05353-t003:** Parameter estimators derived from generalized estimating equation model.

			ß	SE	95% CI Lower	95% CI Upper	*p*
constant			−0.599	0.460	−1.500	0.301	0.192
Gender		female	−0.139	0.044	−0.224	−0.053	0.001
		male	0 ^a^				
BFI		extraversion	−0.065	0.080	−0.221	0.091	0.414
		conscientiousness	−0.001	0.073	−0.144	0.142	0.992
		neuroticism	−0.111	0.094	−0.294	0.072	0.235
		openness	−0.011	0.096	−0.199	0.177	0.906
		agreeableness	0 ^a^				
SAMS factor		Modification	−0.048	0.639	−1.300	1.203	0.940
		Missing Knowledge	2.126	0.592	0.966	3.286	0.000
		Forgetting	0 ^a^				
Education		high	−0.001	0.064	−0.126	0.124	0.986
		middle	−0.048	0.063	−0.173	0.076	0.446
		low	0 ^a^				
Diagnosis		movement disorder	0.064	0.070	−0.074	0.201	0.363
		cerebrovascular disorder	−0.021	0.059	−0.138	0.095	0.721
		epilepsy	0.005	0.096	−0.183	0.192	0.961
		neuromuscular	−0.053	0.060	−0.169	0.064	0.378
		others	0 ^a^				
Living situation		alone	−0.050	0.051	−0.150	0.050	0.332
		not alone	0 ^a^				
Use of nonmedical treatment		no	−0.035	0.047	−0.127	0.058	0.462
		yes	0 ^a^				
Medication change in last 6 months		no	−0.011	.050	−0.109	0.087	0.822
		yes	0 ^a^				
Age			0.002	0.003	−0.005	0.008	0.615
Number of medications/day			0.013	0.012	−0.011	0.036	0.300
BDI			0.016	0.008	0.001	0.031	0.039
HCCQ			−0.041	0.023	−0.087	0.004	0.077
MoCA			0.020	0.015	−0.009	0.049	0.180
TuG			0.010	0.009	−0.007	0.026	0.265
Interactions							
Modification * Number of medications/day			−0.060	0.018	−0.095	−0.025	0.001
Missing Knowledge * Number of medications/day			0.040	0.016	0.008	0.071	0.013
Forgetting * Number of medications/day			0 ^a^				
Modification * BDI			0.026	0.011	0.005	0.048	0.017
Missing Knowledge * BDI			−0.004	0.010	−0.023	0.015	0.692
Forgetting * BDI			0 ^a^				
Modification * MoCA			0.008	0.025	−0.040	0.056	0.755
Missing Knowledge * MoCA			−0.099	0.024	−0.145	−0.053	<0.001
Forgetting * MoCA			0 ^a^				

^a^ Set to 0, since this parameter is redundant. Significant predictors and interactions in bold. BDI II: Beck Depression Inventory II, HCCQ: Health Care Climate Questionnaire, BFI: Big Five Inventory, MoCA: Montreal Cognitive Assessment, TuG: Timed Up and Go test. Dependent variable: factor score. Model: (Constant), Sex, BFI, Factor, Education level, Diagnosis group, Living situation, Use of nonmedical treatment, Medication change in last 6 months, Age, Number of medications/day, BDI. HCCQ, MoCA, Timed Up and Go duration in seconds, Sex * Factor, BFI * Factor, Education Level * Factor, Diagnosis group * Factor, Use of nonmedical treatment * Factor, Medication change * Factor, Living situation * Factor, Age * Factor, Number of medications/day * Factor, BDI * Factor, HCCQ * Factor, MoCA * Factor, TuG * Factor

## Data Availability

The data used in this study is freely available for noncommercial scientific purposes from: Prell, T., & Schönenberg, A. (2022). Data on medication adherence in adults with neurological disorders: The NeuroGerAd study. OSF. doi:10.17605/OSF.IO/KUAPH.

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
