# Peer review of "Adherence to Medication in Neurogeriatric Patients: Insights from the NeuroGerAd Study"

_jcm, 2022, doi:10.3390/jcm11185353_

Round 1

Reviewer 1 Report

The authors have presented an excellent study and statistical information among a few critical disorders patients like stroke, epilepsy... The information presented in this manuscript is very nice.

Though the study has a few critical drawbacks in its presentation:

The study is multivariant and it confuses readers with the conclusions of this work

The sample size is also not consistent throughout the study. The age factor has not been given importance while concluding the results of the present survey.

It will be very important and useful if authors can divide separate diseases and their survey including all parameters. Depression is very common in such neurological disorders as stroke and epilepsy.

The whole manuscript can be divided into specific Neurological Disorders and a similar survey can be a useful output or conclusion for this important study

I recommend reframing the manuscript with a similar study method to induce scientific output.

The English grammar and typo errors need to be corrected throughout the manuscript. 

The title should also be revised before final submission

Author Response

We are immensely grateful to the Reviewer for the overall positive feedback on our manuscript and the constructive comments they provided. Please find our response to the individual comments below:

COMMENT: The study is multivariant and it confuses readers with the conclusions of this work

            RESPONSE: We thank the Reviewer for pointing out this relevant point. As mentioned, our study is complex and thus, it is not possible to include all aspects in the conclusion. However, we added a new Conclusion section and hope that the key findings are now clearer.

COMMENT: The sample size is also not consistent throughout the study.

RESPONSE: Thank you for drawing our eye to this potential inconstistency. As stated in the Methods section, 910 patients participated in the study. Missing data were pairwise treated as stated in the Methods section, which may result in a drop of cases in certain models. We revised the results section to incorporate more detail on the sample size.  

COMMENT: The age factor has not been given importance while concluding the results of the present survey.

RESPONSE: We thank the reviewer for pointing out this oversight of ours. We added a section about the influence of age in our discussion.

COMMENT: It will be very important and useful if authors can divide separate diseases and their survey including all parameters. Depression is very common in such neurological disorders as stroke and epilepsy. The whole manuscript can be divided into specific Neurological Disorders and a similar survey can be a useful output or conclusion for this important study

RESPONSE: We are grateful to the reviewer for bringing up this important topic. In the literature, several analyses have been conducted on nonadherence in individual patient groups, such as Asthma or HIV. Our aim was specifically to assess overarching influential factors in a large cohort, as especially in advancing age where the chances of multimorbidity increase, it is hardly possible to find patients suffering exclusively from one illness. However, we included the Diagnosis variable in all analyses to assess whether the underlying diagnosis itself is a relevant factor predicting adherence when simultaneously considering other relevant parameters such as depression and number of medications. Of note, our analyses revealed that the diagnosis, although influential, was not a key predictor of overall adherence as measured by the SAMS as well as its subfactors (Table 2). Importantly, the diagnosis we used to classify patients is based on the main diagnosis treated during their stay at our hospital, but as many older adults suffer from multimorbidity, they were not exclusively Parkinson’s or Epilepsy patients but may also have suffered from other illnesses. This, as well as the missing clear influence of diagnosis in the elastic net analysis, is why we did not perform separate analyses for each diagnosis group. Likewise, for sub-analyses, sample sizes are relatively low, especially for epilepsy (N = 48), making individual analyses less substantive. Therefore, although sub-analyses can be added if required, we would recommend not to perform them as their informative value may be limited.

COMMENT: I recommend reframing the manuscript with a similar study method to induce scientific output.

RESPONSE: Unfortunately, we do not understand what exactly we should revise or reframe. Could you please provide more detail if this is still a relevant aspect?

COMMENT: The English grammar and typo errors need to be corrected throughout the manuscript. 

RESPONSE: The manuscript was edited by a professional language editing service prior to publication. Additionally, we re-read the manuscript and revised accordingly.

COMMENT: The title should also be revised before final submission

RESPONSE: We reworded the title with respect to the original study protocol: Adherence to medication in neurogeriatric patients: insights from the NeuroGerAd study

Reviewer 2 Report

I would start with commending the authors for providing such a detailed and organized review of medication adherence factors in a large cohort. It has been a very relevant and complex issue and some of the factors are known to medical community but we do not necessarily have too clear evidence. The study validates some of those patient/medication factors (eg. depression, #of meds etc) as a reason for nonadherence. 

The study introduction includes a concise background and problem. Methodology is sound and results in script as well as supplementary material are well reported. I do not see any major flaws in to the study. 

I have few minor suggestions:

- Some of the abbreviations are not described (eg. PCA - I could only find in supplementary material). please review your abbreviations. It would be nice to describe abbreviations as ligands in Figure 1. (eg. DEGS1)

- I will recommend adding a "Conclusion" section summarizing your findings given that discussion is very lengthy (understandaly to describe such complex study). That will send out to clear message to the readers.

Author Response

We are immensely grateful to the Reviewer for providing us with such overall positive feedback on our manuscript. Please find below our response to the constructive suggestions they made:

- Some of the abbreviations are not described (eg. PCA - I could only find in supplementary material). please review your abbreviations. It would be nice to describe abbreviations as ligands in Figure 1. (eg. DEGS1)

            RESPONSE: Thank you for pointing out this oversight on our part, we adjusted the manuscript to provide information on all abbreviations.

- I will recommend adding a "Conclusion" section summarizing your findings given that discussion is very lengthy (understandaly to describe such complex study). That will send out to clear message to the readers

            RESPONSE: We are grateful for this suggestion, as we agree that the study is complex and would benefit from a summary at the end. We added a conclusion section to briefly summarize the key results.

Round 2

Reviewer 1 Report

The Authors have improved a few aspects of the Manuscript after 1st revision request. The English and Grammer still need an eye to interpret a few lines in the discussion and conclusion section to make it more clear.

In The Author's response, the authors of this study have mentioned that the study is complex, I would say the study is interesting but not well framed. The authors have very good strong data but the presentation of the data relative to each variable is very poor and this presentation makes it complex.

The study is having multiple variables that are making the study in multiple directions, For example, if we discuss stroke and post-stroke psychiatric outcomes it's called focused but if we say stroke, post-stroke, Schizophrenia cancer, and Liver disease outcomes are complex.

Although you can revise as per selective variable and re-analyze it. This is a very good study and due to its poor presentation lacking scientific charm into the scientific presentation.

Author Response

Thank you for your speedy response to our revision.
As per the first lines, we re-worked the discussion and conclusion to improve our grammar. We hope that our content is now more understandable to the reader.
Additionally, we re-structured the Methods and Results section to give it a better outline. We also added a summarizing section to the Discussion to round up our complex results. Of note, our data includes patients with four of the most common neurological disease groups. As mentioned before, individual analysis on specific diagnoses already exist in the literature, and dividing our analyses into four subgroups would a) not only make the analysis less substantive due to lower sample size, but also b) make the manuscript even more complex, as all analyses and variables would be included not once but four times. Instead, for this initial manuscript on our large dataset, we aimed to find overarching influential predictors of nonadherence. Therefore, we included patients with different diagnoses, albeit all neurological, and still included the individual diagnosis as a variable to assess its influence. As the dataset is freely available, subgroup analyses can be performed by other researches interested in specific patient groups at a later timepoint. With such a large dataset including various variables, it is simply not feasible to present all possible analyses within one manuscript. Therefore, we hope that our re-structured manuscript is now sufficiently clear in its message for publication.